# Ancient genomes in South Patagonia reveal population movements associated with technological shifts and geography

Nathan Nakatsuka [1,2,20✉], Pierre Luisi [3,20✉], Josefina M. B. Motti [4], Mónica Salemme[5,6], Fernando Santiago [5], Manuel D. D'Angelo del Campo[4,7], Rodrigo J. Vecchi[8], Yolanda Espinosa-Parrilla[9,10], Alfredo Prieto [11], Nicole Adamski[1,12], Ann Marie Lawson[1,12], Thomas K. Harper [13], Brendan J. Culleton[14], Douglas J. Kennett[15], Carles Lalueza-Fox [9], Swapan Mallick[1,12,16], Nadin Rohland[1], Ricardo A. Guichón[4], Graciela S. Cabana [17], Rodrigo Nores [3,18,20✉] & David Reich [1,12,16,19,20✉]

Archaeological research documents major technological shifts among people who have lived in the southern tip of South America (South Patagonia) during the last thirteen millennia, including the development of marine-based economies and changes in tools and raw materials. It has been proposed that movements of people spreading culture and technology propelled some of these shifts, but these hypotheses have not been tested with ancient DNA. Here we report genome-wide data from 20 ancient individuals, and co-analyze it with previously reported data. We reveal that immigration does not explain the appearance of marine adaptations in South Patagonia. We describe partial genetic continuity since ~6600 BP and two later gene flows correlated with technological changes: one between 4700–2000 BP that affected primarily marine-based groups, and a later one impacting all <2000 BP groups. From ~2200–1200 BP, mixture among neighbors resulted in a cline correlated to geographic ordering along the coast.

[1] Department of Genetics, Harvard Medical School, Boston, MA 02115, USA. [2] Harvard-MIT Division of Health Sciences and Technology, Boston, MA 02115, USA. [3] Departamento de Antropología, Facultad de Filosofía y Humanidades, Universidad Nacional de Córdoba, 5000 Córdoba, Argentina. [4] NEIPHPA-CONICET, Facultad de Ciencias Sociales, Universidad Nacional del Centro de la Provincia de Buenos Aires, 7631 Quequén, Argentina. [5] Centro Austral de Investigaciones Científicas (CADIC-CONICET), 9410 Ushuaia, Tierra del Fuego, Argentina. [6] Instituto de Cultura, Sociedad y Estado (ICSE), Universidad Nacional de Tierra del Fuego, 9410 Ushuaia, Tierra del Fuego, Argentina. [7] Laboratorio de Poblaciones del Pasado (LAPP), Departamento de Biología, Facultad de Ciencias, Universidad Autónoma de Madrid (UAM), E-28049 Madrid, Spain. [8] CONICET—Departamento de Humanidades, Universidad Nacional del Sur, 8000 Bahía Blanca, Argentina. [9] Institute of Evolutionary Biology (CSIC-Universitat Pompeu Fabra), 08003 Barcelona, Spain. [10] School of Medicine and Laboratory of Molecular Medicine—LMM, Center for Education, Healthcare and Investigation—CADI, Universidad de Magallanes, Punta Arenas, Chile. [11] Universidad de Magallanes, Avenida Bulnes 01855, Punta Arenas, Chile. [12] Howard Hughes Medical Institute, Harvard Medical School, Boston, MA 02446, USA. [13] Department of Anthropology, The Pennsylvania State University, University Park, PA 16802, USA. [14] Institutes for Energy and the Environment, The Pennsylvania State University, University Park, PA 16802, USA. [15] Department of Anthropology, University of California, Santa Barbara, Santa Barbara, CA 93106, USA. [16] Broad Institute of Harvard and MIT, Cambridge, MA 02142, USA. [17] Molecular Anthropology Laboratories, Department of Anthropology, University of Tennessee, Knoxville, TN 37996, USA. [18] Instituto de Antropología de Córdoba (IDACOR), CONICET, Universidad Nacional de Córdoba, 5000 Córdoba, Argentina. [19] Department of Human Evolutionary Biology, Harvard University, Cambridge, MA 02138, USA. [20]These authors contributed equally: Nathan Nakatsuka, Pierre Luisi, Rodrigo Nores, David Reich. ✉email: nathan_nakatsuka@hms.harvard.edu; pierre.luisi@unc.edu.ar; rodrigonores@ffyh.unc.edu.ar; reich@genetics.med.harvard.edu

South Patagonia, defined here as the region south of the 49th parallel in South America (Fig. 1), has been occupied by humans since at least the time of the ~12600 bp Tres Arroyos rockshelter on Isla Grande de Tierra del Fuego (calendar years before present; all dates calibrated with marine reservoir effect correction in what follows)[1]. A handful of sites date to the early (~13000–8500 bp) and middle (~8500–3500 bp) Holocene, and site density increased considerably in the Late Holocene (<3500 bp)[2]. During this span, archaeological research has provided evidence of multiple material culture shifts that could potentially have been associated with movements of people[3].

The earliest shift relates to seafaring technology, including adoption by at least ~6700 bp of canoes and harpoons, which made possible the hunting of sea lions and other pinnipeds even in seasons when they were not available on the shore, allowing the settlement of nomadic hunter-gathering populations in the archipelagos[4,5]. The development of this technology has been hypothesized to reflect either in situ origin from land hunter–gatherers, or spread of techniques from the north via copying of ideas or movement of people[1,6,7].

The second shift occurred in the Western Archipelagos and involved changes in raw material and shape of tools, with green obsidian (probably sourced from the Otway Sound in the South of the Western Archipelago) as a characteristic marker of the first period[8] (~6700–6300 bp), and large bifacial lithic projectile points of different materials in the later period (~5500–3100 bp)[5,8]. The disruption in green obsidian use has been hypothesized to reflect a loss of cultural knowledge about the location of the source of this raw material, potentially due to arrival of new people unfamiliar with the landscape[8].

The third shift involved changes across the whole region by ~2000 bp with evidence of population growth and technological innovations. The archaeological record of the Beagle Channel shows diversification in lithic projectile point designs, which has been interpreted as change, reorganization, and broadening in prey procurement strategies[9]. In the northern part of Tierra del Fuego the use of boleadoras (stone spheres bound with rope used as throwing weapons) ceased by ~1500 bp[10]. In addition, a new type of pedunculated lithic projectile point used as a head for projectile weapons appeared by ~2000 bp. Its size reduction around ~900 bp was associated with the appearance of bow and arrow technology[11,12]. The similarity of these Late Holocene projectile points with those from historical times documents an element of cultural continuity at least from ~2000 bp[3], although this does not prove genetic continuity as techniques can be copied, and similar environments can lead to parallel innovations.

When Europeans arrived in the 16th century, they described five Native groups in South Patagonia (Fig. 1) practicing two broad subsistence strategies optimized for different terrains: the plateaus and lowlands of the east and north vs. the irregular coast with islands and archipelagos in the west and south. Terrestrial hunter–gatherers included the Aónikenk (or Southern Tehuelche), who extended along the eastern slope of the mainland, and the Selk'nam (or Ona), who occupied the north of the island of Tierra del Fuego. These two groups relied primarily on hunting guanaco and birds, and gathering shellfish from the seashore[13]. The Yámana (or Yaghan) in the Beagle Channel region and the Kawéskar (or Kawésqar or Alacalufe) in the Western Archipelago (including the Otway Sound and Strait of Magellan shores) had a high reliance on marine resources that could easily be accessed by sea canoes. Finally, the Haush (or Mánekenk) of the southeastern tip of the island of Tierra del Fuego on the Mitre Peninsula did not have navigation technology, but archaeological evidence indicates that they hunted both terrestrial and marine prey[13–16]. The relationships between the five groups have been the subject of debate, with some arguing that mating among different groups was common in boundary areas[17], and others suggesting that such unions were rare[18] (see Supplementary Note 1 for more details).

Genome-wide studies can provide direct information about whether or not movements of people accompanied changes evident in the archaeological record. Uniparental markers analysis showed that South Patagonians have only C and D mitochondrial haplogroups and low Y-chromosome diversity, consistent with a bottleneck in the founding groups followed by strong genetic drift and isolation[19–22]. de la Fuente et al. published the first genome-wide data from four individuals dated to around 1000 bp along with data from 61 modern Patagonians[19]. The authors demonstrated a substantial degree of continuity from the archaeological individuals to present-day ones. The terrestrial Selk'nam shared alleles at an equal rate with the maritime Kawéskar and Yámana to the limit of the statistical resolution of that study[19]. Another study published two individuals of ~6600 and 4700 bp, and showed they were more closely related to 1000 bp individuals and to historical groups from the region than to any ancient or modern groups in other parts of South America, documenting more than 7000 years of detectable shared ancestry in South Patagonia[23].

Several questions remain to be addressed (referenced throughout the paper): was there genetic continuity across time in the region or is there any detectable population change correlating with (1) marine diet specialization at ~6700 bp, (2) technological shifts such as the abandonment of green obsidian use between ~5500 and 3100 bp, and/or (3) the transition from the use of boleadoras to pedunculated lithic points ~2000 bp? (4) What was the extent of gene flow among neighboring South Patagonian groups? (5) Were the inhabitants of Mitre Peninsula genetically more similar to maritime or terrestrial groups? (6) How do the ancient groups relate to the ones after European contact?

In the present study, we report new genome-wide data for 19 individuals in South Patagonia from ~5800 to 100 bp, including the first such data from the Mitre Peninsula and inland at the south of the continent (Fig. 1); we also report a ~2400 bp individual from the Pampas in Argentina (archaeological information is provided in Supplementary Note 2). Compared with previous ancient DNA data from the region (6 pre-European contact[19,23] and 11 post-contact[24]), our data fill in spatio-temporal gaps, particularly in the east and north of Tierra del Fuego.

## Results

**Community engagement and ethics.** During more than 30 years of archaeological work in the region, one of the authors (R.A.G.) held numerous meetings with different members of present-day Patagonian communities living in the same geographic areas where the ancient samples were located. Reaching spaces for dialogue and joint learning between members of the Native and scientific communities has been a central theme since the beginning of this research and allowed exchanging perspectives on the objectives of bioanthropological studies. In addition, on several occasions, R.A.G. organized and carried out educational activities in schools at different educational levels in the cities of Río Grande and Ushuaia, Tierra del Fuego (Argentina). Some members of Native communities expressed interest in establishing through genetic analyses how the ancient human skeletal remains found by chance and conserved in the museums relate to present-day people. This study followed that interest, and to facilitate the distribution and understanding of our findings, we translated the abstract (Supplementary Note 3) and main conclusions into Spanish (Supplementary Note 4) and shared them with members of the Native communities.

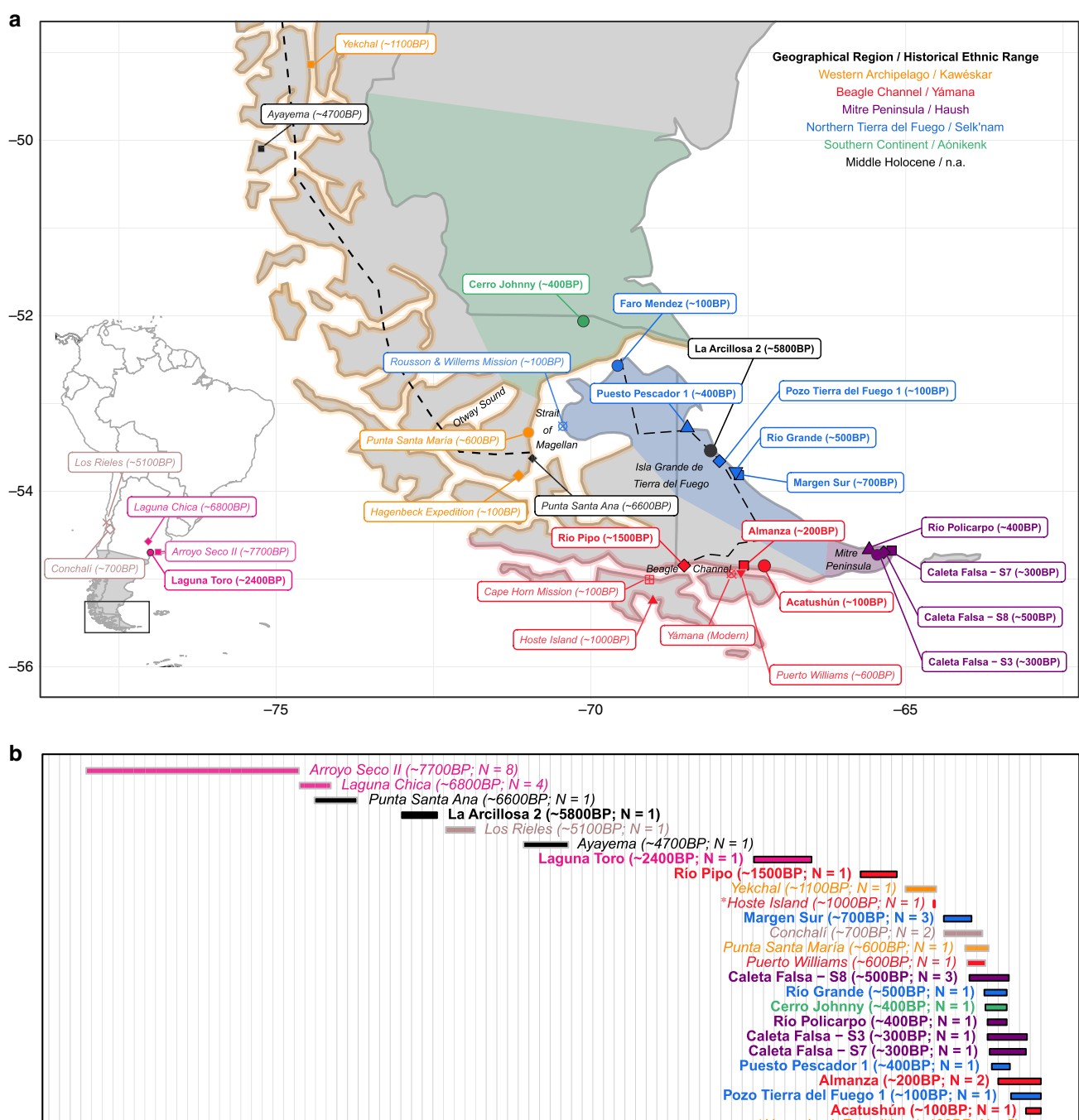

**Fig. 1 Geographic and temporal distribution.** Newly and previously reported data are in bold and italics, respectively; color coding is in the legend. **a** Geography: we used site coordinates or reported location, except for Raghavan et al.[24] samples that were geographically reassigned according to historical evidence. The dashed lines represent routes of movement used to calculate plausible migration distances. The continuous line marks the border between Argentina in the east and Chile in the west. Inset: location of South Patagonia (rectangle) and the broader Patagonia region (following McCulloch et al.'s[76] definition; gray), along with the locations of ancient individuals mentioned in the main text but falling outside the range of the main map. The historical ranges of groups were adapted from Borrero[7]. The map was generated in R using the maps, ggplot2, ggrepel, and dplyr packages to get the map, plot it, label it, and provide accents, respectively. **b** Time ranges (number of individuals per site in parentheses). Sites for which radiocarbon dates were not available are labeled with an asterisk. Dates were calibrated for the Southern hemisphere and corrected for maritime reservoir effect (see "Methods").

**Authenticity of ancient DNA**. We verified the authenticity of the analyzed data based on all samples meeting the following criteria: (1) a rate of cytosine-to-thymine changes at the ends of the aligned fragments of >3%[25], (2) mitochondrial DNA (mtDNA) contamination point estimates below 5%[26], (3) X-chromosome contamination point estimates in males below 3%[27], and (4) genome-wide contamination[28] point estimates below 5%. No individuals were removed based on these analyses. We report but do not analyze one individual (I12365) who was found to be the brother of I12367. Supplementary Data 1 includes details on all the ancient individuals we analyzed.

**Uniparental markers, population size estimates, and variants of phenotypic relevance**. All mitochondrial haplotypes of the South Patagonians were C or D, reflecting the only haplogroups found in the Fuegian archipelago to date, with higher rates of D1g5 and C1c in Northern Tierra del Fuego, C1b in the Beagle Channel, and co-occurrence of C1b and D1g5 in the Mitre Peninsula (Supplementary Data 1). D1g5 is a widespread clade in ancient and modern people from Argentina and Chile[29,30], and probably differentiated in the early stages of the Southern Cone colonization, since it has geographically structured internal clades[29]. We also observed one individual in North Tierra del Fuego from Late Holocene period who was D4h3a, which today is concentrated along the Pacific coast of both South and North America[31]. All Y-chromosomes fall into the Q1a2a haplogroup. Moreover, Q1a2a1a was observed in all the individuals with sufficient genetic information for more specific paternal lineage determination, except for one (Q1a2a1b; in Mitre Peninsula from the Late Holocene period), a similar skew to that seen across South America today.

We performed conditional heterozygosity analyses and found that ancient Patagonian groups had rates of variation at polymorphic sites[32] as low as the groups in the world with the lowest variation today (Supplementary Fig. 1). This suggests persistent low population size, consistent with previous inferences based on uniparental marker analyses[19–22]. We were not able to determine the date of the population bottlenecks given our lack of high coverage whole-genome sequencing data from the ancient individuals[33].

We examined the status of the analyzed individuals for several previously reported variants associated with cold tolerance. However, the small sample sizes were insufficient to allow us to document significant allele frequency change over time and thus we were not able to carry out formal tests for natural selection (Supplementary Data 2).

**Correlation of genetic ancestry with geography and language**. We detect a significant degree of continuity in South Patagonia since at least 6600 bp, as symmetry $f_4$-statistics show that the earliest Patagonians share more alleles with later Patagonians relative to Pampas, Argentina (*Argentina_ArroyoSeco2_7700BP* or *Argentina_LagunaChica_6800BP*) or Central Chile (*Chile_-LosRieles_5100BP*)[34] (Supplementary Data 3A). We also analyzed all individuals with unsupervised ADMIXTURE (Supplementary Fig. 2), principal components analysis (PCA) (Supplementary Fig. 3), an $F_{ST}$-based heatmap (Supplementary Fig. 4), and measurements of shared genetic drift between pairs of individuals using statistics of the form $f_3(Mbuti; Ind1, Ind2)$ (Fig. 2). A multidimensional scaling (MDS) plot of $f_3$-statistics-based matrix shows that Middle Holocene individuals are distinct from the Late Holocene individuals (Fig. 2b), with the important exception of *Chile_Ayeyama_4700BP*, which shows a slight shift toward later Western Archipelago individuals, a signal that reflects an important genetic event that we discuss in detail in what follows.

In the Late Holocene, the genetic structure of South Patagonia correlated with geography, diet/technology, and linguistic group, with largely separated clusters in the Beagle Channel region, Western Archipelago, and Southern Continent/North Tierra del Fuego. However, there are also gradients, with individuals from the Mitre Peninsula forming a cline between the North Tierra del Fuego/Southern Continent and Beagle Channel individuals; and the modern Yámana individual lying between ancient individuals from the Western Archipelago and Beagle Channel (Fig. 2).

We correlated pairwise genetic drift distances to geographical, temporal, and linguistic distances (based on historically attested languages), as well as distances based on differences in subsistence resources (Table 1; "Methods" and Supplementary Data 4). Mantel tests were significant for distances based on all four variables (P values based on 10,000 permutations < 0.0002). When performing partial Mantel tests controlling for the other variables to determine if each variable had additional explanatory power beyond the others, the association remained significant for language (P = 0.0004) and geography (P = 0.0183); we observed qualitatively similar findings when performing these analyses in other ways (Supplementary Data 5).

Based on the clear evidence for correlation of genetics with geography and post-European contact language family, we named the Late Holocene groups in each region according to the ethnic groups recognized at the time of European contact. Thus, we refer to individuals from the Western Archipelago as Kawéskar, the Beagle Channel area as Yámana, the Mitre Peninsula as Haush, the North Tierra del Fuego island as Selk'nam, and the South Continent as Aónikenk. We recognize that this is an over-simplification, because we cannot know how the individuals from ~1500 to 500 bp self-identified, if language differentiation reflected the languages in historical periods, if the cultures of the past were similar enough to those of historical periods to have a meaningful degree of continuity, or if there were further important subdivisions beyond the groupings recognized by Europeans at the time of contact[35]. Categorization into five discrete groups also masks substructure and differentiation within groups (e.g., the cline in ancestry we observe in the Haush of differential relatedness to the Yámana on the one hand and the Selk'nam on the other). However, we use these names because the genetic data do not contradict the traditional terms and indeed correlate to them strongly.

**Genetic differentiation that distinguishes Late Holocene maritime and terrestrial groups appeared by ~4700 bp**. When we computed $f_4$-statistics comparing the oldest individuals to more recent individuals (Supplementary Data 3B), we observed that *Chile_PuntaSantaAna_6600BP* and *Argentina_LaArcillosa2_5800BP* were equally distant genetically to later groups (including both maritime and terrestrial-reliant groups). Moreover, *Chile_PuntaSantaAna_6600BP* and *Argentina_LaArcillosa2_5800BP* were also equally distant genetically to all American groups outside the region at any time transect that we could analyze. Based on archaeological and isotope data[36] (Supplementary Fig. 5), the Western Archipelago *Chile_PuntaSantaAna_6600BP* individual is one of the first known South Patagonians with a primarily marine-based diet (also described for the *Chile_Ayayema_4700BP* individual[23]); while the North Tierra del Fuego *Argentina_LaArcillosa2_5800BP* individual had a primarily terrestrial diet[37]. This addresses our first question: our results suggest that the first appearance of maritime adaptation in South Patagonia is not explained by an immigration event from the north.

However, the Late Holocene Kawéskar and Yámana groups are significantly more related (|Z| > 3) to *Chile_Ayayema_4700BP*

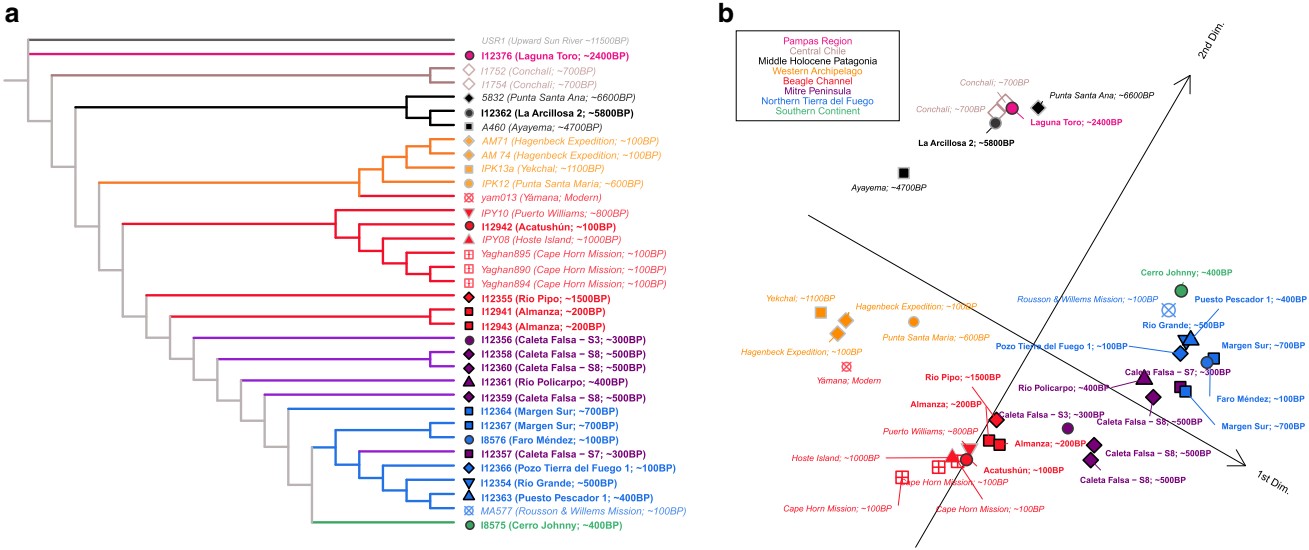

**Fig. 2 Population structure in South Patagonia. a** Neighbor-joining tree created using the matrix of inverted statistics $(f_3(Mbuti; Ind1, Ind2))^{-1}$, with an ancient Beringian[77] as an outgroup[72, 73]. **b** Multidimensional scaling (MDS) plot of the matrix of statistics $1-f_3(Mbuti; Ind1, Ind2)$. The matrix of the first two dimensions of MDS was rotated 30 degrees to emphasize the striking geographic correlation of the genetic cline of Late Holocene samples to the coastline. Only individuals with >100,000 SNPs were included; newly and previously reported data are in bold and italics, respectively.

**Table 1 Correlation of genetic distances with relevant variables.**

| Variable | Simple Mantel test: R (P value) | Partial Mantel test: P value |
|---|---|---|
| Geography | 1.90E − 01 (2.00E − 04) | 1.83E − 02 |
| Diet/technology | 6.62E − 02 (4.00E − 04) | 9.75E − 01 |
| Language | 2.32E − 01 (<1E − 4) | 4.00E − 04 |
| Time | 2.31E − 02 (<1E − 4) | 2.58E − 01 |

P values are based on 10,000 permutations. Multiple $R^2$ for partial Mantel test: $R^2 = 0.30931$.

than to *Chile_PuntaSantaAna_6600BP* or *Argentina_LaArcillosa2_5800BP* (Supplementary Data 3B), consistent with the pattern evident in Fig. 2b. The fact that the ancient Selk'nam, Aónikenk, or Haush did not show significant ($|Z| < 1.5$) affinity for *Chile_Ayayema_4700BP* suggests that the ancestry present in *Chile_Ayayema_4700BP* made a larger contribution to later groups that relied mainly on marine resources accessible from sea canoes than to eastern groups like Selk'nam and likely Aónikenk that relied mainly on terrestrial resources. This persisted to historical times as *Chile_Ayayema_4700BP* shares more alleles ($Z = 3.5$) with *Yamana_CapeHorneMission_Grouped_100BP* than with *Selknam_RoussonandWillemsMission_100BP* (Supplementary Data 3C). Therefore, in contrast to the archaeological evidence for a shift to marine specialization, the shift away from green obsidian use in the Western Archipelagos between ~5500 and 3100 bp does have a correlate in our genetic findings (our second question), as it occurred during the time of the marine-adapted *Chile_Ayayema_4700BP* individual who bears significant additional affinity to Late Holocene people from this region[23]. The fact that this individual, who is from the most northern part of South Patagonia, shows specific genetic affinity to later marine-adapted groups in South Patagonia is consistent with population change during this time frame. Specifically, it suggests gene flows connecting marine-adapted groups throughout South Patagonia (although we are not able to determine the direction of such gene flows from our genetic analysis).

All groups outside of Patagonia were symmetrically related to these earliest Patagonian groups to the limits of our resolution (Supplementary Data 3B), consistent with all these changes being due to local developments in the southern tip of South America, albeit with important movements within this broad region.

**Gene flow from North Patagonia into South Patagonia in the middle to Late Holocene.** To test for genetic interaction between Patagonia and other regions in America after ~4700 bp (Question 3), we tested for asymmetry between Late Holocene (~1500–100 bp) vs. Middle Holocene (over ~4700 bp) Patagonian groups compared to other Native Americans, assessing if statistics of the form $f_4$(Mbuti, OtherSouthAmericans; MiddleHolocenePatagonians, LateHolocenePatagonians) were significantly different from 0. The only consistently significant signal was an excess allele sharing of *Chile_Conchali_700BP* from Central Chile far to the north with some of the later groups (Aónikenk, Haush, Yámana, and Selk'nam) relative to the Middle Holocene individuals (Supplementary Data 3D). There is no evidence this is due to South Patagonian gene flow into Central Chile, as statistics like $f_4$(Outgroup, SouthPatagoniaAfter4700BP; *Chile_LosRieles_5100BP*, *Chile_Conchali_700BP*) were all consistent with 0 (Supplementary Data 3E). We obtained further support for this direction of gene flow when we used *qpAdm* to attempt to model *Chile_Conchali_700BP* as a mixture of *Chile_LosRieles_5100BP* and any Late Holocene Patagonian group; *Chile_Conchali_700BP* is always modeled as consistent with having no Late Holocene Patagonian ancestry (Supplementary Data 6A), providing little scope for a scenario of large-scale South Patagonian ancestry moving northward into Central Chile.

Using *qpAdm*[38] to model the Late Holocene South Patagonian groups, we found that the maritime Kawéskar and Yámana could be modeled as 45–65% (±4–7%; we use 1 standard error to report *qpAdm* estimations) *Chile_Conchali_700BP*-related ancestry and the rest *Chile_Ayayema_4700BP*-related (Supplementary Data 6A, all models pass at $P > 0.02$ where $P$ values are estimated by block jackknife resampling). These models work with *Chile_PuntaSantaAna_6600BP* and *Argentina_LaArcillosa2_5800BP* among the outgroups in *qpAdm*; in contrast, when

*Chile_PuntaSantaAna_6600BP* or *Argentina_LaArcillosa2_5800BP* were used as the second source instead of *Chile_Ayayema_4700BP* (used as an outgroup), the models do not fit ($P < 0.005$). These results suggest little if any direct continuity from 6600 to 5800 bp groups to Late Holocene maritime groups in South Patagonia, consistent with substantial re-peopling of South Patagonia archipelagos in the Middle Holocene. In contrast, the eastern Selk'nam could not be modeled with *Chile_Ayayema_4700BP*-related ancestry ($P < 0.005$) and instead only fit ($P > 0.02$) as a mixture of ~50–60% *Chile_Conchali_700BP*-related and the rest *Argentina_LaArcillosa2_5800BP* or *Chile_PuntaSantaAna_6600BP*-related ancestry. The Haush fit as ~50–60% *Chile_Conchali_700BP*-related and the rest as any Middle Holocene groups (we do not have resolution to resolve the source). The Aónikenk had a borderline fit ($0.005 < P < 0.015$) with ~50–60% ($\pm 6$–7%) *Chile_Conchali_700BP* and either *Chile_PuntaSantaAna_6600BP* or *Chile_Ayayema_4700BP* (*Argentina_LaArcillosa2_5800BP* did not fit). Additional analyses (below) suggest that Aónikenk has an ancestry most similar to that of Selk'nam, and so we favor the model of *Chile_Conchali_700BP* and *Chile_PuntaSantaAna_6600BP*, which works for both groups.

In summary, all our working *qpAdm* models for the Late Holocene South Patagonians involve a mixture of about half ancestry from a group related to *Chile_Conchali_700BP*, and about half ancestry from one Mid-Holocene South Patagonian lineage (*Chile_PuntaSantaAna_6600BP* or *Chile_Ayayema_4700BP*) that we have sampled in the studied dataset and that diverged from each other at least by ~6600 bp (the date of *Chile_PuntaSantaAna_6600BP*). This could be explained by north-to-south gene flow of *Chile_Conchali_700BP*-related ancestry into South Patagonia admixing into each of the divergent groups across the region, but cannot be explained by gene flow in the reverse direction, which would be expected to cause *Chile_Conchali_700BP* to be modeled as having ancestry from either the *Chile_PuntaSantaAna_6600BP*-related or *Chile_Ayayema_4700BP*-related lineage, which is not supported by $f_4$-statistics, *qpAdm*, or *qpGraph* modeling (Supplementary Data 3E and Fig. 3).

Taken together, our analyses thus suggest at least three major north-to-south gene flows affecting South Patagonia: the first bringing *Chile_PuntaSantaAna_6600BP* ancestry by at least the date of this individual, the second bringing *Chile_Ayayema_4700BP* ancestry into the archipelagos of South Patagonia at least by ~2000 bp (the average date of formation-by-admixture of the Late Holocene ancestry cline, see below), and the third bringing *Chile_Conchali_700BP* ancestry into all South Patagonia again by at least ~2000 bp.

We did not detect excess allele sharing of genetic affinity of groups outside Patagonia (such as the *Argentina_LagunaToro_2400BP* or present-day Chane individuals) with the Selk'nam or Aónikenk relative to Kawéskar or Yámana (Supplementary Data 3F). However, our reference data are sparse, and a particular weakness is that we lack data from individuals from further south (between the Pampas and South Patagonia regions) that could plausibly have interacted genetically with the Aónikenk and Selk'nam. Future ancient DNA sampling could allow one to test if such groups exchanged genes in the Mid-to-Late Holocene with people in South Patagonia. We did not find ancestry from groups differentially related to non-Americans in any of the individuals (also called Population Y ancestry, Supplementary Data 3G), consistent with previous analyses of individuals from South Patagonia[23].

**Genetic mixtures between geographically neighboring South Patagonian groups**. To obtain insight into the extent of genetic isolation among the Late Holocene Patagonian groups (Question 4), we computed symmetry $f_4$-statistics.

The Selk'nam were genetically intermediate between their neighbors (Fig. 1a), as the Aónikenk to their north shared more alleles with them than with the Haush and Yámana to their east and south; similarly, the Haush and Yámana shared more alleles with the Selk'nam than with the Aónikenk (Supplementary Data 3H). Accordingly, we used *qpAdm* to model the Selk'nam as $63.8 \pm 9.2\%$ Aónikenk related and 36.2% Yámana related (Supplementary Data 6B). Using *DATES* (54), which studies the breakdown of allele covariance in a target group relative to two source populations, we infer an average admixture date of $1902 \pm 282$ years ago (we use 1 standard error to report *DATES* results), assuming a generation time of 28.5 years (Supplementary Data 6C).

The Haush too were genetically intermediate between their neighbors, with Yámana attracting Haush relative to Selk'nam and Selk'nam attracting Haush relative to Yámana (Supplementary Data 3H). We confirmed directly that the Haush are admixed (Question 5) through a significantly negative ($Z = -6.6$) statistic of the form $f_3$(Haush; Yámana, Selk'nam) (Supplementary Data 3I). The MDS plot suggests a cline of Selk'nam- and Yámana-related ancestry in the different Haush individuals, so we used *qpAdm* to model the ancestry of each individual separately. We estimated that they vary from 10.2 to 44.8% Yámana related (Supplementary Data 6C). The fact that there are substantial ancestry differences among the Haush indicates that mixing between the groups may have been actively occurring in the period we sampled. We estimate an average admixture date of $1334 \pm 171$ years ago.

The Yámana were also genetically intermediate between their neighbors, with the Selk'nam attracting the Yámana relative to the Kawéskar and the Kawéskar attracting the Yámana relative to the Selk'nam (Supplementary Data 3H). This signal was not detected in a past study[19] which reported Selk'nam as consistent with being equally related to Kawéskar and Yámana. However, that study relied on a single ~100 bp Selk'nam individual (which we confirm has symmetric relationship to Kawéskar and Yámana to the limits of the resolution of the statistics), while our additional data analyze many Selk'nam individuals and leverage this larger dataset to successfully detect asymmetry. No significantly negative $f_3$-statistic unambiguously demonstrated admixture (Supplementary Data 3I), but this could be due to lack of power or, alternatively, genetic drift in the ancestors of the Yámana since admixture, which can mask an admixture signal. We could model Yámana as $54.2 \pm 14.4\%$ Kawéskar related and 44.2% Selk'nam related (one individual had $83.3 \pm 16.7\%$ Kawéskar-related ancestry, but the others were between 51 and 56%). With *DATES* we determined the admixture date to be $1627 \pm 313$ years ago (the absolute inferred dates in the past are similar even when only the older Yámana individuals are used).

Taken together, these results show that there was active mixture between South Patagonian groups ~2200–1200 years ago with a cline ranging from the Aónikenk on one end to the Kawéskar on the other, and that gene flow slowed since that time (if it had continued at that rate we would not see an older date for the more recent individuals). The recent reduction of gene flow suggests the possibility that cultural differentiation became greater in the more recent period.

**Admixture graph model**. We used *qpGraph* to fit an admixture graph to the data and to model the relationships of the different South Patagonian groups to each other and to selected other South American groups (Fig. 3). The model fit captures many of the individual findings of this study. The Pampas group *Argentina_LagunaToro_2400BP* is equally related to all South Patagonians. The lineage of *Chile_Ayayema_4700BP* is modeled as

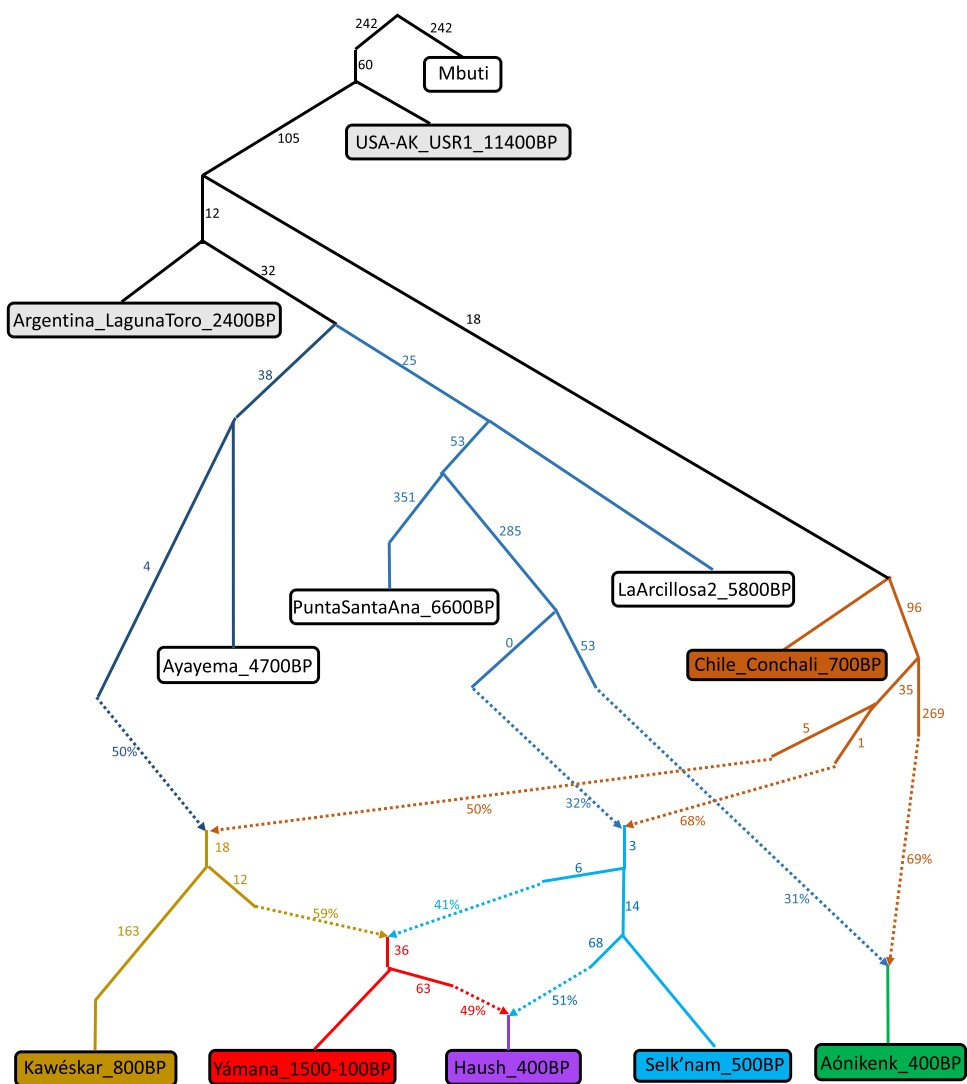

**Fig. 3 Admixture graph model summarizing key findings.** Maximum |$Z$-score| = 2.6 for a difference between observed and expected $f$-statistics (|$Z$| = 2.7 restricting the analysis to transversions). The model presented fits only after adding small proportions of deeply diverging ancestry into *PuntaSantaAna_6600BP* and *Kawéskar_800BP* (splitting before the radiation of Native Americans), which we hypothesize reflects not real ancestry but rather technical artifacts due to these samples being shotgun sequenced and not UDG treated, causing them to be attracted to the outgroup (without modeling these edges, shown in Supplementary Fig. 6, the maximum |$Z$-score| is 5.1, but this drops to 3.3 with only transversions). Dashed lines indicate admixture between two different lineages with percentages being the admixture proportions. Numbers on solid lines are genetic drift with units of $F_{ST}$ × 1000. $Z$-scores were determined from standard errors obtained from jackknife resampling.

contributing to the maritime groups *Yamana_1500-100BP* and *Kaweskar_800BP* but not to other South Patagonians, reflecting the fact that genetic variation specific to later maritime groups had developed by ~4700 bp. Since *Chile_Ayayema_4700BP* is from western South Patagonia but the other earlier Middle Holocene Patagonians are not, this implies a migration related to *Chile_Ayayema_4700BP* displacing the earlier lineages. The model captures our inferences that South Patagonians after ~4700 bp have additional ancestry from a source related to the Central Chilean *Chile_Conchali_700BP* reflecting Late Holocene major north-to-south gene flow (Question 3). The model also reflects the cline of ancestry as mixtures of each other with Kawéskar at one extreme and Aónikenk at the other extreme (Question 4). Finally, the model confirms that *Haush_400BP*, the group with a mosaic of cultural traits shared with terrestrial and maritime groups, can be modeled as a mixture between *Selknam_500BP* (terrestrial adaptation) and *Yamana_1500-100BP* (maritime adaptation) (Question 5). Without these key admixture events in

the model, we could not find a graph that fit (the maximum |$Z$|-score between observed and expected statistics in the fitting graph is over 3.5).

**Modern individuals are most related to ancient individuals from the same region**. We compared modern Yámana and Kawéskar[19,39,40] to ancient Patagonians (Question 6), and found significant excess allele sharing of modern individuals from each regional grouping with the members of the ancient group from the same region (Supplementary Data 3J), consistent with previous findings[19]. We extended these findings to the Chono, Chilote, and Huilliche, who live just north of Kawéskar, with whom they are genetically most similar. We also co-analyzed an additional dataset of modern Patagonian groups[19,40] with the ancient groups in an admixture graph[19]. We could model the modern Yámana, Kawéskar, Huilliche, and Pehuenche (a group just north of Huilliche) as a mixture of European ancestry

(reflecting post-colonial admixture), local pre-contact Native American ancestry, and Central Chile (*Chile_Conchali_700BP*)-related ancestry (Supplementary Fig. 6; we did not attempt to fit such models for Chono and Chilote due to the scarcity of SNPs). We observe a decreasing gradient of Central Chile-related ancestry from Pehuenche, Huilliche, Kawéskar, to Yámana (ordered in line with their geographic distances from *Chile_Conchali_700BP*). Thus, the ancient DNA data are capturing only the southern part of a geographic cline in *Chile_Conchali_700BP*-related ancestry in Patagonia. Future ancient DNA sampling could provide additional details on the origin and timing of the development of this cline.

## Conclusions

Our results falsify the hypothesis that the earliest marine adaptation in South Patagonia was due to a large-scale immigration into South Patagonia of people from the north who were already using this economic strategy (Question 1 in the introduction). Instead, local people adopted the technology or invented it independently. However, our results indicate the arrival of a later stream of people from the northwest (following the stream that brought the ancestors of the mid-Holocene individuals, potentially the initial colonization event), which brought ancestry from a lineage related to the maritime-associated *Ayayema* (~4700 bp) individual, replacing the lineages related to *Punta Santa Ana* (~6600 bp) and *La Arcillosa2* (~5800 bp) that were previously established in South Patagonia itself. The arrival of this new stream of people could be related to the change in lithic technology between ~5500 and 3100 bp, characterized by the interruption of green obsidian use and the introduction of large biface projectile points in the Western Archipelago and Beagle Channel regions[3,8] (Question 2). In addition, a third source of ancestry from Central Chile spread between ~4700 and 2000 bp. This could be related to different processes that occurred in the Late Holocene, such as the increase in site density as a sign of population growth, and the cessation of use of *boleadoras* replaced by new hunting technologies that emerged since ~2000 marked by the use of pedunculated lithic projectile points as heads of throwing weapons[3] (lances and arrows) (Question 3). The shared linguistic family between North, Central, and South Patagonia groups in historical and modern times[41] could be related to this signal.

In the Late Holocene, we detect gene flow among neighbors especially from 2200 to 1200 years ago and attenuating afterward (Question 4). A plausible scenario is that the Haush adopted some of their maritime and terrestrial adaptations from the people with whom they exchanged genes (Question 5), as the genetic data demonstrates that they were socially connected through exchange of mates in this time. The Haush spoke a language in the same family (Chon) as the Selk'nam and Aónikenk, while the Yámana language is an isolate or related to Kawéskar[42], but there was nevertheless gene flow across these linguistic boundaries. Finally, population continuity in South Patagonia after European contact (Question 6) is supported by the genetic affinity of modern Yámana and Kawéskar with ancient individuals from their respective regions.

We did not find evidence of genetic exchange with Argentinian groups outside Patagonia (based on lack of affinity to the Pampas individual *Argentina_LagunaToro_2400BP* or present-day Chane). However, we do find evidence of large-scale movements of people from Central Chile and within Patagonia over thousands of years. An important goal for further research should be to carry out additional ancient DNA sampling not only in South (especially on the western coast) but also in Central and North Patagonia where our analysis of modern populations

detects a cline of Central Chilean-related ancestry reflecting north–south gene flow, to provide higher resolution and additional insights into the interactions among people that shaped the Native cultures of this unique region of the world.

## Methods

**Ethical approval**. We performed this study following ethical guidelines for working with human remains, treating them with respect due to deceased people. The ancient skeletal samples we analyzed were all curated at the Museo del Fin del Mundo (Ushuaia, Argentina), the Centro Austral de Investigaciones Científicas (Ushuaia, Argentina), the Universidad Nacional del Sur (Bahía Blanca, Argentina), or the Universidad de Magallanes (Punta Arenas, Chile). Samples of the skeletal material were exported with full Argentinean and Chilean governmental permissions (see Supplementary Data 7 for details).

**Direct AMS $^{14}$C bone dates**. We report 15 direct AMS $^{14}$C dates on bone and teeth for 14 ancient individuals (Supplementary Data 1); information about sample processing methodology is in Supplementary Note 5.

**Calibration of radiocarbon dates**. All calibrated $^{14}$C ages were calculated using OxCal version 4.3[43], using differing mixtures of the southern hemisphere terrestrial (SHCal13[44]) and the marine (Marine13[45]) calibration curves. Marine dietary contribution was estimated using stable carbon and nitrogen isotope measurements from collagen (Supplementary Data 1). Nitrogen provides a benchmark for the relative importance of marine dietary resources, with $\delta^{15}N$ values of ~11.5‰ indicating a wholly terrestrial diet and ~22.0‰ indicating a predominately (~90%) marine diet. We delineated five categories of calibration curve mixing, assuming marine-derived diets of 0%, 20%, 40%, 60%, and 80%, respectively (Fig. S5A), assuming an uncertainty value of ±10%. Probability distributions are shown in Fig. S5B. Observed stable isotope distributions group by region and agree with the known subsistence strategies of the Kawéskar, Yámana, Haush, Selk'nam, and Aónikenk. We used a marine reservoir correction (ΔR value) of 221 ± 40 (1 standard deviation) from Puerto Natales, Chile[46].

**Ancient DNA work**. Tooth powder was obtained in dedicated clean rooms at the University of Tennessee, Knoxville using a freezer mill for 18 individuals and at Harvard Medical School by drilling for 2 individuals. DNA extraction for all samples was performed using a method optimized to retain small DNA fragments either manually[47,48] or with an automated liquid handler using silica-coated magnetic beads[49]. We prepared double-stranded Illumina sequencing libraries, pretreating with the enzyme uracil-DNA glycosylase (UDG) to minimize analytical artifacts due to the characteristic cytosine-to-thymine errors in ancient DNA[25], using an automated liquid handler and substituting the MinElute columns used for cleaning up reactions with silica-coated magnetic beads and buffer PB (Qiagen), and the MinElute column-based PCR cleanup at the end of library preparation with SPRI beads[50,51]. We enriched the libraries for sequences that overlapped both mtDNA[52] and about 1.24 million nuclear targets for two rounds of enrichment[38,53,54], either independently (1240k and MT separately) or together (1240kplus). We sequenced the enriched products on an Illumina NextSeq500 using v.2 150 cycle kits for 2 × 76 cycles and 2 × 7 cycles to read the indices. Skeletal material from all 20 ancient individuals screened for this project yielded usable DNA data.

**Computational processing of initial sequence data**. We used two different data processing methods, which have been shown to produce negligible differences in inferences about population history[55]. Supplementary Data 1 specifies which individuals were processed using each method.

For method 1, we merged paired forward and reverse reads that overlapped by at least 15 nucleotides using SeqPrep (https://github.com/jstjohn/SeqPrep), and used the highest quality base to represent each nucleotide. We aligned the sequences to the human genome reference sequence (GRCh37, hg19) and the reference sequence (MT RSRS) using the *samse* command of *BWA* (version 0.6.1)[56] with parameters: $n = 0.01$, $o = 2$, $l = 16500$. We removed duplicates using Picard MarkDuplicates (http://broadinstitute.github.io/picard/), requiring matching indices and barcodes to declare duplicates.

For method 2, we merged paired forward and reverse reads that overlapped by at least 15 nucleotides using custom software (https://github.com/DReichLab/ADNA-Tools). We allowed one base mismatch when the forward and reverse bases both had quality at least 20 and up to three mismatches when the base read quality was <20, and retained the higher quality base in the case of a conflict. We restricted to merged sequences of at least 30 base pairs. We aligned FASTQ files using the *BWA* (version 0.7.15-r1140) *samse* command[56] with parameters: $n = 0.01$, $o = 2$, $l = 16500$ to the hg19 human reference and the MT RSRS. We removed duplicates with Picard.

For both methods, we removed two nucleotides from the end of each sequence for partial UDG-treated samples and ten nucleotides for UDG-untreated samples (from previously published studies). We selected a single sequence at each site covered by at least one sequence to represent the individual's genotype.

**Contamination estimation**. We determined whether the data were consistent with authentic ancient DNA by measuring the damage rate in the first nucleotide, flagging individuals as potentially contaminated if they have a <3% cytosine-to-thymine substitution rate in the first nucleotide for a UDG-treated library and <10% substitution rate for a non-UDG-treated library as assessed using PMD tools[57]. To estimate mitochondrial contamination, we used *contamMix* version 1.0–12[26], running the software with down-sampling to 50X for samples above that coverage. We used ANGSD to determine evidence of contamination in males based on polymorphisms on the X chromosome[27] with the parameters minimum base quality = 20, minimum mapping quality = 30, bases to clip for damage = 2, and all other parameters set to default. We also measured contamination in the autosomal DNA (chromosomes 1–22) of both males and females using a tool based on breakdown of linkage disequilibrium[28]. All samples passed quality control (individual I12941 had a 1.8% deamination rate, but we did not remove this individual because all other estimates showed negligible contamination, and low deamination is not so surprising for relatively recent samples (this individual dates to ~200 calbp)).

**Kinship analyses**. To determine genetic kinship within our dataset, we analyzed all pairs of individuals at non-CpG autosomal sites and computed an average mismatch rate at all SNPs covered by at least one sequence read for both individuals assessed, then compared these rates to rates from known kinship relationships[58]. We removed I12365 from the main analysis dataset as we genetically detected him to be a brother of I12367 (higher coverage), but we report the data fully.

**Present-day human data**. We used present-day human data from the Simons Genome Diversity Project[59], as well as data from 78 Native Americans genotyped on the Axiom LAT1 array[19,40], and a whole-genome shotgun sequence of a present-day Yámana (yam013)[19].

**Y-chromosome and mitochondrial DNA analyses**. For Y-chromosome haplogroup determination, we used a modified version of yHaplo (https://github.com/23andMe/yhaplo) designed to work with ancient DNA, determining the most derived mutation for each individual using the tree of the International Society of Genetic Genealogy and considering the presence of upstream mutations consistent with the assigned Y-chromosome haplogroup using Yfitter version 0.3[60].

For mtDNA haplogroup determination, we generated VCFs using Samtools[61] version 1.10 with the parameters (minimum mapping quality 30, minimum base quality 20). We ran Haplogrep[62] phylotree version 17 to attain a haplogroup assignment (Supplementary Data 1).

**Calculation of distances for different explanatory variables**. We calculated distance metrics for a variety of variables.

For temporal distances between two samples, we calculated the absolute difference between the average of the 95.4% date range in calbp, defining setting to 0 the date for modern samples.

For subsistence resources, we assumed that the distance between individuals who based their subsistence primarily on terrestrial resources (Northern Tierra del Fuego and Southern Continent) and individuals for whom maritime resources were most important (Western Archipelago and Beagle Channel) was 2. For individuals for whom both resources were common (Mitre Peninsula), the distance to individuals with unique primary resources was set to 1. The distance between individuals who used similar food procurement strategies was set to 0.

For linguistic group, we took into account the fact that Selk'nam/Ona, Haush, and Tehuelche/Aónikenk are Chonan languages, and that the former are more similar to one another than to the latter. Accordingly, the distance between an individual belonging either to Mitre Peninsula or Northern Tierra del Fuego was set to 1, while the distance for individuals from one of these two regions to individuals from the Southern Continent was set to 2. Considering that Yámana/Yaghan and Káweskar are language isolates, we set to 4 the distance from individuals from either the Western Archipelago or the Beagle Channel to individuals from the three remaining regions. However, these two languages may be more similar to each other than to Chonan languages[41,42]; therefore, we set to 3 the distance between individuals from the Western Archipelago and individuals from the Beagle Channel. The distance between individuals from the same geographic region was set to 0.

For geographic distance, we had to contend with the complexity of the potential routes for movement of people in the area. We conjectured that a coastal route between the Mitre Peninsula and the Beagle Channel or North Tierra del Fuego was the most probable. We also considered the possibility that moving from North Tierra del Fuego to the Beagle Channel might have followed an interior route that roughly corresponds to the present-day National Road 3 (dashed line in Main Fig. 1a). The routes connecting the Cerro Johnny site located at the South of the Continent to Tierra del Fuego sites were computed as described for Northern Tierra del Fuego, adding the distance between this site and its closest projection on the Northern Tierra del Fuego coast. The distance between the Cerro Johnny site and the southernmost Western Archipelago sites was calculated following the continental coast. Yámana and Káweskar people used canoes; therefore, we considered the shortest maritime routes connecting the southern part of the Western Archipelago to the Beagle Channel. The distances between the

southernmost Western Archipelago and North Tierra del Fuego and Mitre Peninsula were calculated as the sum of the length of the shortest maritime path between continental and Tierra del Fuego coasts with the length of the shortest terrestrial path in Tierra del Fuego. We also hypothesized that moving south in the Western Archipelago could be simplified by following a direct route (see dashed line in main Fig. 1a). We thus computed the distance from the Yekchal site to the others by the sum of this route to the coast close to the Punta Santa Ana site, and then by the paths described for southwestern Archipelago sites. The distance between individuals from the same site was set to 0, while the length of a straight line connecting two sites was used for any pair of sites from the same geographical region. The estimates were performed using the *geor* package in R[63] and the map from the *maps* package with resolution parameter 0 (as shown in Fig. 1).

All the estimated distances are available in Supplementary Data 4.

**Testing association of genetic distances with variables of interest**. We tested if genetic distance between pairs of individuals was associated with Linguistic, Temporal, Geographical and/or Subsistence distances. Pairwise genetic distances were set to either $1-f_3(Mbuti; Ind1, Ind2)$ or $(f_3(Mbuti; Ind1, Ind2))^{-1}$. We performed simple Spearman correlation and linear regression analyses. To correct for relationships among the explanatory variables, we also performed partial Spearman correlation and partial linear regression analyses, first correcting the genetic distances for the three other explanatory variables through a multivariate linear regression. For each coefficient estimate, we computed a 95% confidence interval (±1.96 standard error) with a weighted block jackknife over 5-Mb blocks[64]. Finally, we performed simple (including the matrix for only one explanatory variable) and partial Mantel tests (including the distance matrices for the four explanatory variables) using the multi.Mantel function in R with 10,000 permutations of the genetic distance matrix. In all of these analyses, we assumed that each individual was independent even though this is not strictly true due to shared genetic drift within groups.

**Grouping of individuals**. All individuals were first analyzed separately for the outgroup-$f_3$-based MDS and neighbor-joining tree. For some *qpAdm* and *DATES* analyses, the individuals were then grouped by region (Western Archipelago, Beagle Channel, Mitre Peninsula, North of Tierra del Fuego island, and South Continent), sequencing method (capture or shotgun) and age as in the Fig. 1a color-coding scheme. In general, we name groups using the following nomenclature: *Region_SiteName_Age BP*[65]. Age BP of a group comprised of more than one individual is computed by averaging the mean of the estimated date range (Supplementary Data 1).

**Conditional heterozygosity analyses**. Conditional heterozygosity is an estimate of genetic diversity in a group obtained by sampling a random allele from each of two randomly chosen individuals at a known panel of polymorphisms[32]. We performed these analyses for transversion variants on all South American groups with at least two individuals per site using *POPSTATS* (https://github.com/pontussk/popstats) with the September 26, 2018 version with default settings. We computed this on samples from this study, ancient South Americans from Brazil[66], Central Chile[66], the Andes[23,40,66], the Pampas region in Argentina[66], and Patagonia[19,23,24], and on present-day Native American human sequencing data[40]. We restricted to individuals without substantial European admixture (inferred from the ADMIXTURE analyses below).

**ADMIXTURE analysis**. We merged the genotype data used for Condition Heterozygosity Analyses (but including transition sites) with Axiom LAT1 genotyping data for present-day Native Americans[19,40], as well as 2 × 15 randomly sampled individuals from Italy and Spain from the Phase 3 of 1000 Genomes Project[67]. We removed ambiguous genotypes (A/T, C/G), and SNPs and individuals with more than 50 and 90% of missing genotypes. We filtered out variants with minor allele frequency <1% and pruned to remove linkage disequilibrium (–indep pairwise flag with 50 SNP windows, 5 SNP steps, and 0.5 $r^2$ threshold in PLINK2). We ended with 106,285 SNPs for 116 modern Native South American individuals, 72 ancient individuals, and 30 South European reference individuals. We ran unsupervised ADMIXTURE[68] version 1.3.0 with ten replicates for each K, reporting the replicate with the highest likelihood. We show results for K = 2–7 in Supplementary Fig. 2.

**Masking out regions of non-native ancestry in admixed individuals**. We merged the 116 modern Native South American individuals with 503 European, 504 African, and 347 American individuals from 1000 Genomes Project Phase 3[67]. After removing SNPs with more than 2% of missing genotypes and minor allele frequency below 1% and individuals with more than 10% missing genotypes, we ended with 129,269 SNPs and 1462 individuals. After LD-pruning (–indep pairwise flag with 50 SNP windows, 5 SNP steps, and 0.5 $r^2$ threshold in PLINK2), we ran unsupervised ADMIXTURE with K = 3 to estimate European, African, and Native American ancestry. Individuals with <99% Native American ancestry were considered as admixed, while the others were set as Native American reference. We ran RFMIXv2[69] with 503, 504, and 69 European, African, and Native American reference samples, respectively, to identify the genomic regions with Native American ancestry in the remaining 387 admixed individuals. RFMIX was run with

the following settings: we used the –n 5 flag to reduce bias, we set the number of expectation–minimization iterations to 2 (–em 2 flag) to improve the local ancestry calls and set the –reanalyze-reference flag to leverage the Native American haplotypes segregating in admixed individuals, and we set both –c and –s flags to 0.2 (corresponding to the -w 0.2 flag in RFMIX v1[69]). RFMIX requires the genotype data to be pre-phased, and this was done using shapeIT2[70] with default parameters and using the reference haplotypes for 2,504 worldwide individuals from the 1000 Genomes Project. We also used the average genetic map provided by the 1000 Genomes Project. For a given allele at a given SNP for a given individual, if the maximum posterior probability of a given ancestry was >0.9, the allele was assigned to that ancestry, otherwise it remained with unknown ancestry. We checked that the local ancestry inference procedure was consistent with global ancestry analyses and observed that the Native ancestry proportions estimated globally (through ADMIXTURE) or locally (through RFMIXv2) have a Spearman correlation coefficient of 0.9988, with a maximum difference of 0.04. For each South American individual, we performed masking to only keep the regions that are inferred to be Native American on both chromosomes.

**Principal components analysis and $F_{ST}$ analyses.** We merged the masked genotype data for the 108 modern Native South American individuals to the South American ancient samples described above. We removed SNPs and individuals with more than 50% and 90% of missing genotypes in the compiled genotype data, respectively, obtaining genotype data for 106,981 SNPs, and 101 and 72 modern and ancient individuals, respectively. We performed PCA with *smartpca*[71], and used the default parameters except inbreed: YES, lsqproject: YES, and turning off outlier removal. We show results for PC2 vs. PC1 in Supplementary Fig. 3. We also used *smartpca* to compute $F_{ST}$ values between all groups that had at least two individuals. For this analysis we used fstonly: YES and inbreed: YES with all the other settings left at default. We show results within a heatmap for pairwise $F_{ST}$ used as a similarity index with hierarchical clustering-based dendrogram reordering in Supplementary Fig. 4.

**Symmetry statistics and admixture tests (*f*-statistics).** We used the *qp3pop* and *qpDstat* packages in ADMIXTOOLS[34] version 6.0 to compute $f_3$-statistics and $f_4$-statistics (using the f4Mode: YES parameter in *qpDstat*) with standard errors computed with a weighted block jackknife over 5-Mb blocks. We used the inbreed: YES parameter to compute $f_3$-statistics to account for our random allele choice at each position. We computed outgroup $f_3$-statistics of the form $f_3(Mbuti; Pop1, Pop2)$, which measures the shared genetic drift between population 1 and population 2. We created a matrix of the outgroup-$f_3$ values between all pairs of populations. We converted these values to distances by subtracting the values from 1 and generating an MDS plot in R. We converted the original values to distances by taking the inverse of the values and generating a neighbor-joining tree using PHYLIP version 3.696's[72] neighbor function and setting *USA-AK_USR1_11400BP* as the outgroup. We displayed the tree using Itol and set all of the tree lengths to ignore[73].

**Admixture graph modeling.** We used *qpGraph*[74], removing transition SNPs at CpG sites and using default settings with outpop: Mbuti.DG and useallsnps: YES. We used the 1240k dataset and created a modified graph of *Mbuti.DG, USA-AK_USR1_11400BP, Chile_Conchali_700BP,* and *Argentina_LagunaToro_2400BP,* and then successively added in additional populations in all combinations allowing up to one admixture from the existing groups in the graph. We took the graph with the lowest maximum Z-score and then repeated the process, adding another population until all populations of interest were added. We first started with the two oldest individuals (*Chile_PuntaSantaAna_6600BP* and *Argentina_LaArcillosa2_5800BP*). We then added the groups in order: *Chile_Ayeyama_4700BP, Kaweskar_WesternArchipelago_Grouped_800BP, Selknam_NorthTierradelFuego_Grouped_500BP, Yamana_BeagleChannel_Grouped_1500-100BP, Aonikenk_CerroJohnny_400BP,* and *Haush_MitrePeninsula_Grouped_400BP.* We added additional deep-rooting admixture edges for *Kaweskar_WesternArchipelago_Grouped_800BP* and *Chile_PuntaSantaAna_6600BP,* because they were shotgun sequenced and processed differently which plausibly explains their (likely artifactual) attraction to Mbuti. We merged the data with the Axiom LAT1 unmasked genotype set of modern Patagonian individuals[19,40] and added in the present-day Yámana, Kawéskar, Huilliche, and Pehuenche from that dataset, manually attempting to find the best fit with an extra admixture edge added to account for recent European admixture.

**Modeling of ancestry proportions.** We used *qpAdm*[38] in ADMIXTOOLS version 6.0 to estimate the proportions of ancestry in the different Late Holocene individuals. We analyzed each group as a mixture of the two groups geographically closest to them (except we never used Haush as a source population due to their genetic heterogeneity). We also used *qpAdm* to formally model the Late Holocene individuals as mixes of the Early and Middle Holocene individuals and *Chile_Conchali_700BP.* For these analyses, we modeled each of the Late Holocene individuals as a mix of one of the Early or Middle Holocene individuals (*Chile_PuntaSantaAna_6600BP, Chile_Ayayema_4700BP,* or *Argentina_LaArcillosa2_5800BP*) and *Chile_Conchali_700BP* with the outgroups *Chane_modern, Peru_Cuncaicha_900BP, Argentina_LagunaToro_2400BP,*

*Chile_LosRieles_10900BP, Chile_LosRieles_5100BP, Argentina_ArroyoSeco2_7700BP, Argentina_LagunaChica_6800BP,* and the other two Early and Middle Holocene Patagonia individuals. *P* values were obtained by jackknife resampling and a likelihood ratio test (two sided). We considered models to fit if $P > 0.02$; $P < 0.005$ failed; and models with $0.005 < P < 0.02$ borderline (Supplementary Data 6).

**Admixture dating analyses.** We used *DATES* version 1510[75], which estimates the age of admixture in ancient DNA samples based on breakdown of allelic covariance over genetic distance in the target group relative to two source populations. We used the default settings with jackknife: YES and analyzed each group as a mixture of the two groups adjacent to them (except for Yámana, which we analyzed as a mixture of Kawéskar and Selk'nam rather than Haush due to the genetic heterogeneity in Haush).

**Analyses of phenotypically relevant SNPs.** We examined SNPs known to be relevant to phenotypic traits[66] as well as additional ones with evidence of modulating cold tolerance in humans (Supplementary Data 2). We used *samtools* version 1.3.1[61] *mpileup* with the settings -d 8000 -B -q30 -Q30 to obtain information about each read from the bam files of our samples. We used the fasta file from human genome GRCh37 (hg19) for the pileup. We counted the number of derived and ancestral variants at each analyzed position using a custom Python script.

**Reporting summary.** Further information on research design is available in the Nature Research Reporting Summary linked to this article.

## Data availability
All sequencing data are available from the European Nucleotide Archive, accession number: PRJEB39010. Genotype data obtained by random sampling of sequences at ~1.24 million analyzed positions are available at the Reich lab website: https://reich.hms.harvard.edu/datasets.

## Code availability
Code for the software used in this paper is provided at the following locations: SeqPrep version 1.2 (https://github.com/jstjohn/SeqPrep), custom software for merging reads (https://github.com/DReichLab/ADNA-Tools), BWA version 0.6.1 (bio-bwa.sourceforge.net), Picard version 2.23.0 (http://broadinstitute.github.io/picard/), OxCal version 4.3 (https://c14.arch.ox.ac.uk/oxcal.html), ContamMix version 1.0–12 (https://github.com/DReichLab/ADNA-Tools), ANGSD version 0.930 (https://github.com/ANGSD/angsd), ContamLD version 1.0 (https://github.com/nathan-nakatsuka/ContamLD), yHaplo (https://github.com/23andMe/yhaplo), Samtools version 1.10 (http://samtools.sourceforge.net/), Haplogrep version 17 (http://haplogrep.uibk.ac.at/index.html), ADMIXTURE version 1.3.0 (https://www.genetics.ucla.edu/software/admixture/download.html), RFMIX version 2 (https://github.com/slowkoni/rfmix), shapeIT2 version 2 (https://jmarchini.org/shapeit2/), EIGENSOFT version 5.0 (https://github.com/DReichLab/EIG), ADMIXTOOLS version 6.0 (https://github.com/DReichLab/AdmixTools), DATES version 1510 (https://github.com/priyamoorjani/DATES), POPSTATS September 26, 2018 version (https://github.com/pontussk/popstats), PMD tools version 0.60 (https://github.com/pontussk/PMDtools), multi.Mantel function in R phytools version 0.6–60 package (https://cran.r-project.org/web/packages/phytools/index.html), geor package version 1.8 (https://cran.r-project.org/web/packages/geoR/index.html), maps package version 3.3.0 (https://cran.r-project.org/web/packages/maps/index.html), ggplot2 package version 3.3.2 (https://cran.r-project.org/web/packages/ggplot2/index.html), ggrepel package version 0.8.2 (https://cran.r-project.org/web/packages/ggrepel/index.html), dplyr package version 1.00 (https://cran.r-project.org/web/packages/dplyr/index.html), Yfitter version 0.3 (https://sourceforge.net/p/yfitter/wiki/Home/), PLINK2 version 2.0 alpha (https://www.cog-genomics.org/plink/2.0/), a custom script for kinship determination based on mismatch rates as described in Kennett et al.[58] (available upon request but not yet ready for broader distribution), and custom scripts for outgroup-$f_3$ neighbor-joining tree and MDS plot plotting as well as allele counting (https://github.com/nathan-nakatsuka/Patagonia).

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

## Acknowledgements

We are grateful to the members of Patagonian Native communities whom we consulted in the course of our research, in particular the Selk'nam, Yagán (Yámana), and the Mapuche-Tehuelche. We thank the Museo del Fin del Mundo, the Centro Austral de Investigaciones Científicas, the Universidad Nacional del Sur, and the Universidad de Magallanes for allowing us to access their collections. We thank Jakob Sedig, Mark Lipson, Matthew Mah, Iosif Lazaridis, and Iñigo Olalde for critical comments and helpful discussions. We thank Miguel Vilar for logistic support and Ricardo A. Verdugo for sharing modern Patagonian genotype data. N.N. is supported by a NIGMS (GM007753) fellowship. R.N. was supported by a National Geographic Society grant in the pilot program "Ancient DNA: Peopling of the Americas, 2018" and by CONICET (PIP 2015-11220150100953CO, PUE 2016 IDACOR, and BecExt 2017). J.M.B.M. was supported by ANPCyT (PICT 2015-1405). Archaeological research in Argentina was funded by grants to M.S. (CONICET PIP 0422/10 and 6199 and ANPCyT 05-38096) and to F.S. (CONICET PIP 0302). D.R. was supported by National Institutes of Health grant GM100233, by an Allen Discovery Center grant, and by grant 61220 from the John Templeton Foundation; D.R. is also an investigator of the Howard Hughes Medical Institute. C.L.F. was supported by the grant PGC2018-095931-B-100 (MCIU/FEDER, UE). R.N., J.M.B.M., R.A.G., M.S., F.S., and R.J.V. are members of CONICET, Argentina.

## Author contributions

J.M.B.M., M.S., F.S., M.D.D.C., R.J.V., Y.E.P., A.P., C.L.F., and R.A.G. collected and described archaeological material and site contexts. N.A., A.M.L., N.R., G.S.C., and R.N. performed or supervised sample preparation. N.R., N.A., and A.M.L. generated genetic data. T.K.H., B.J.C., and D.J.K. performed or supervised A.M.S. radiocarbon dating analysis and marine correction. S.M. performed bioinformatics analyses. N.N. and P.L. performed population genetics analyses. N.N., P.L., J.M.B.M., R.N., and D.R. interpreted the data. J.M.B.M., R.A.G., N.N., P.L., R.N., and D.R. conceived the study. N.N., P.L., J.M.B.M., R.N., and D.R. wrote the paper with the assistance of other co-authors. R.N. and D.R. co-directed the study.

## Competing interests

P.L. provides consulting services to myDNAmap S.L. The remaining authors declare no competing interests.
