## [Peer Review File · Nature Communications]

Reviewers' comments:

Reviewer #1 (Remarks to the Author):

This manuscript reports of the analysis of the genomes of ancient human remains from Southern Patagonia and compares the results of these analyses to archaeological, linguistic and modern genetic data. The authors seek to answer five interrelated questions: Is there any detectable population change correlating with 1) marine diet specialization at ~6750 BP, 2) technological shifts, such as the abandonment of green obsidian use, replaced by local raw materials between ~5500-3100 BP, or 3) the transition from the use of boleadoras to the use of pedunculated points ~2000BP? 4) What was the extent of gene flow among neighboring South Patagonian groups? 5) Were the inhabitants of Mitre Peninsula genetically more similar to maritime or terrestrial groups? 6) How do the ancient groups relate to the ones after European contact? These questions are answered for the most part and in this sense, the manuscript represents a valuable contribution to the regional scientific literature.

There are, however, a number of the aspects of the manuscript that require clarification so that readers from beyond the relatively small number of regional specialists in Patagonian/Southern Chilean and Argentine archaeology can better understand the significance of the research. Most of these points are fairly "small" but should be addressed in a revision.

- 1) Line 75: the Monte Verde site is not typically described in the literature as being Patagonian. The authors may be technically correct that the site falls into that geographic definition but that is not common usage.
- 2) The authors describe the subsistence adaptations of the Kaweska and Yamana peoples as "marine" oriented but never really define what that means in this context. They need to be explicit about the type of maritime focus of these groups and the degree to which they exploited near-shore or other marine life. In fact, each of the other groups in the broader region exploited marine life—shellfish, for example. Variability in these subsistence systems should be defined so readers can understand the magnitude of the changes argued by the authors.
- 3) The significance of "green obsidian" is remarked upon as signaling a technological shift. The location of the source is never identified in the manuscript. Is this resource closer to one group or another? It is said to have a genetic correlate but what is the presumed archaeological settlement pattern of the people who were its consumers? Instead of simply asserting a shift, the authors should relate it to landuse patterns practiced by the ancient peoples of the region.
- 4) Line 607, Fig, S2: the authors compare their aDNA results to a number of ancient and modern samples across South America. I am curious, however, as to why they did not use the SMP sample from Lindo et al (2018) as part of this exercise. The date of this sample falls between the Cuncaicha and earliest Lauricocha sample and before the latest ones. Is there a sound justification for this?
- 5) Biological sex of individuals is listed in the site descriptions (lines 732-1004). Was sex information recovered from the analysis of aDNA? If so, it would have been useful to have it presented. In the same regard, a table of the C and N isotope data would be valuable so that other researchers could more easily evaluate the data themselves than to mine it from the site descriptions.
- 6) Lines 559-595, conclusions: These could be more clearly stated by referring back to the original research questions found on lines 177-183. As it stands, the reader has to tack back and forth from the conclusions to the questions to see more clearly the degree to which they were answered by the research.
- 7) I am pleased to see that there was consultation with the Indigenous peoples of the region (lines 225-237). However, the description was also rather vague. Are there organizations of Indigenous peoples who could have been consulted? A better description of the consultative process would be welcome.

Reviewer #2 (Remarks to the Author):

This is a very interesting manuscript that provides a refreshing overview of the population history of Southern Patagonia. There are only a few suggestions or questions:

1. Use of Aónikenk. While there is usually more than one known name for referring to the native groups from Patagonia, the one that present-day communities use the most or prefer should be used. In this particular case is Tehuelche, but along the text the authors used Aónikenk. Is there any justification for this? Otherwise, please change by Tehuelche through the manuscript.
2. The proper spelling of the northwest maritime group should be Kawésqar (q instead of k).
3. The Ethics Statement is a great section. Only two comments: a) Please include the corresponding permits in the supplementary material; b) the authors suggested some work has been done associated with the present-day communities, but it is not clear if they actually support this particular research. Beyond the governmental permissions, is there any statement from the communities to this research? Please enclosed, clarify or add in the acknowledgment section.
4. Regarding samples: did you screen a larger set of samples in order to produce these new 20 genomes? Please include this information in the Material/Method section. This information can contribute to the understanding of the DNA preservation and feasibility of paleogenomic research in the region. There is a report of a modern Yámana sample (yam013), but the reference is incorrect since the referred paper does seem to have whole-genome shotgun sequencing of present-day individuals (only array data)
5. Lines 329-330. Please consider using a different expression for referring to ethnographic names. These are not just labels.
6. Section "Gene Flow from Northern Patagonia into Southern Patagonia in the Mid to Late Holocene": a) According to the SI, the f4 results seem to be consistent with Chile_Conchali_700BP having an excess allele sharing with later groups only 50% of the time (14 out of 30 cases, counting also the groups). Is this really a consistently significant signal? Please include a sentence with this fact in the section or justify the use of "consistently significant signal"; b) How can be Chile_Conchali_700BP be modeled (individuals from outside Patagonia). Can this contribute to have a deeper understanding of the source of this new ancestry? c) Does this gene flow correlate with previous inferred events described by Posth et al., 2018 and/or Moreno-Mayar et al., 2018? (maybe not in the time, but in the sort of ancestry/signal)
7. Figure 3: the model is based on the results of the f4 test. From the individuals used, it seems that Kawésqar_800BP doesn't have a significant excess of allele sharing with Conchali, compared to earlier individuals. How this affects the model? Can the model be replicated using: a) any other Kawésqar individual or b) removing transitions (to decrease the technical artifact mentioned in the figure). Are these results consistent with the current model?
8. Software DATES. Why didn't you use the software for estimating the earlier gene flows? (Conchali for example). Is there a technical limitation? The radiocarbon date of the samples provides a limit, but having an additional estimate can contribute to a better understanding of the process.
9. Line 563: "arrival of a later stream of people". Does it involve the arrival of new people or, as the authors suggested, involve the genetic differentiation of maritime groups maintaining gene flow between them (with lower rates in the last millennium)? Please rephrase the sentence considering that population changes cannot only be explained by the arrival of new people.
10. The authors briefly mentioned population sizes inferred by the density of archaeological sites. Does the data allow any population size estimates? (using MSMC for example).

Reviewer #3 (Remarks to the Author):

This is a very interesting paper and the authors are very thorough with the population genetic analyses utilized to form their conclusions. My main comments refer to the structure of the

manuscript, including a missed opportunity to highlight how these samples relate to other ancient genomes available from South America. I am also concerned that many of the interesting results have been buried in the supplement. That being said, here are my suggested edits:

1. The abstract reads like a results section. I would suggest rewording to a summary fashion. The first two sentences are especially descriptive.

2. For the introduction, while I appreciate the in-depth coverage of the cultures, the authors may want to summarize with some key differences between the five populations and move the rest to the supplement.

2. I was especially pleased to see that the ethics statement was included in the main manuscript. However, the description of where the samples are from should be sent to the supplement and perhaps keep the statement about community engagement with indigenous communities in the manuscript.

3. On the other hand, some of the analyses performed are quite interesting but they are buried in the supplement. For examples, the comparisons between linguistic, geographic, and temporal distances with genetic drift seems very interesting and novel. Perhaps the authors could visualize the results in some manner and put it in the main manuscript. Conversely, I don't particularly find Figure 2 to be informative or interesting.

3. It is strange to be referring to the ancient samples by historical ethnic groups for all of the reasons mentioned by the authors. Wouldn't it be less problematic to refer to the samples by the region from which they were unearthed?

4. It seems that the authors draw many conclusions from the F4 analyses. I would recommend weighing all of the evidence from the plethora of analyses done before drawing such hefty conclusions.

5. I also think that at least one of the analyses with regard to continental gene flow should be visualized and featured in the manuscript. For example, was the admixture graph tried with other available genomes from South America? I think this would be a very interesting result to see. It would also be nice to see the continental level analysis presented in a PCA or structure plots. I realize that this is a regional paper but given that such few ancient genomes are available for South America, I believe that considering the samples in a larger scope would increase the impact of the paper.

Reviewers' comments:

Reviewer #1 (Remarks to the Author):

This manuscript reports of the analysis of the genomes of ancient human remains from Southern Patagonia and compares the results of these analyses to archaeological, linguistic and modern genetic data. The authors seek to answer five interrelated questions: Is there any detectable population change correlating with 1) marine diet specialization at ~6750 BP, 2) technological shifts, such as the abandonment of green obsidian use, replaced by local raw materials between ~5500-3100 BP, or 3) the transition from the use of boleadoras to the use of pedunculated points ~2000BP? 4) What was the extent of gene flow among neighboring South Patagonian groups? 5) Were the inhabitants of Mitre Peninsula genetically more similar to maritime or terrestrial groups? 6) How do the ancient groups relate to the ones after European contact? These questions are answered for the most part and in this sense, the manuscript represents a valuable contribution to the regional scientific literature.

There are, however, a number of the aspects of the manuscript that require clarification so that readers from beyond the relatively small number of regional specialists in Patagonian/Southern Chilean and Argentine archaeology can better understand the significance of the research. Most of these points are fairly “small” but should be addressed in a revision.

1) Line 75: the Monte Verde site is not typically described in the literature as being Patagonian. The authors may be technically correct that the site falls into that geographic definition but that is not common usage.

The sentence was removed.

In addition, we are now explicit about the definition of the geographic region of Patagonia in the legend of Figure 1. We used McCulloch et al. (1997) definition for Patagonia “located between latitude 39° and 55° in South America”. We agree with the referee that which region Patagonia referred to was not clear in the manuscript, so we clarified it.

McCulloch, R., Clapperton, C., Rabassa, J., and Carrant, A. P. (1997). The natural setting. The glacial and post-glacial environmental history of Fuego-Patagonia. In McEwan, C., Borrero, L. A., and Prieto, A. (eds.), Patagonia. Natural History, Prehistory and Ethnography at the Uttermost End of the Earth, British Museum Press, London.

2) The authors describe the subsistence adaptations of the Kaweska and Yamana peoples as “marine” oriented but never really define what that means in this context. They need to be explicit about the type of maritime focus of these groups and the degree to which they exploited near-shore or other marine life. In fact, each of the other groups in the broader region exploited marine life—shellfish, for example. Variability in these subsistence systems should be defined so readers can understand the magnitude of the changes argued by the authors.

We agree this was not clear. We now emphasize canoe use and access to resources not available on the shore as our definition of marine economy. For example:

Lines 74-77: “The earliest shift relates to seafaring technology including adoption by at least ~6700 BP of canoes and harpoons which made possible the hunting of sea lions and other pinnipeds even in seasons when they were not available on the shore, allowing the settlement of nomadic hunter-gathering populations in the archipelagos^{5,6,7}.”

Lines 101-107: “The Yámana (or Yaghan) in the Beagle Channel region and the Kawéskar (or Kawésqar or Alacalufe) in the Western Archipelago (including the Otway Sound and Strait of Magellan shores) had a high reliance on marine resources that could easily be accessed by sea canoes. Finally, the Haush (or Mánekenk) of the southeastern tip of the island of Tierra del Fuego on the Mitre Peninsula did not have navigation technology, but archaeological evidence indicates that they hunted both terrestrial and marine prey^{16,17,18,19}.”

Lines 949-950: “terrestrial hunter-gatherers (Selk’nam/Haush) compared to sea canoeists (Yámana and Kawéskar).”

3) The significance of “green obsidian” is remarked upon as signaling a technological shift. The location of the source is never identified in the manuscript. Is this resource closer to one group or another? It is said to have a genetic correlate but what is the presumed archaeological settlement pattern of the people who were its consumers? Instead of simply asserting a shift, the authors should relate it to land use patterns practiced by the ancient peoples of the region.

We included what is known about the source of green obsidian in the Western Archipelagos and how the change in raw materials is interpreted.

Lines 80-84: “The second shift occurred in the Western Archipelagos and involved changes in raw material and shape of tools, with green obsidian (probably sourced from the Otway Sound in the South of the Western Archipelago) as a characteristic marker of the first period.”

Lines 84-86: “The disruption in green obsidian use has been hypothesized to reflect a loss of cultural knowledge about location of the source of this raw material, potentially due to arrival of new people unfamiliar with the landscape¹².”

4) Line 607, Fig, S2: the authors compare their aDNA results to a number of ancient and modern samples across South America. I am curious, however, as to why they did not use the SMP sample from Lindo et al (2018) as part of this exercise. The date of this sample falls between the Cuncaicha and earliest Lauricocha sample and before the latest ones. Is there a sound justification for this?

The SMP sample from Lindo et al., 2018, is unfortunately of relatively low coverage (~73,100 SNPs covered on the 1240K array) and has European ancestry contamination by our analysis. The low coverage is exacerbated when merging with data from present-day individuals, leaving only 7,732 SNPs to be analyzed, which is too low for our analysis.

5) Biological sex of individuals is listed in the site descriptions (lines 732-1004). Was sex information recovered from the analysis of aDNA? If so, it would have been useful to have it presented. In the same regard, a table of the C and N isotope data would be valuable so that other

researchers could more easily evaluate the data themselves than to mine it from the site descriptions.

The genetically determined sex and C/N isotope values are given in Online Table 1. We now make this more clear:

Lines 1012-1013: “A summary of all information obtained for genetic or radiocarbon dating, including genetic sex and carbon and nitrogen isotope data is presented in Supplementary Online Table 1.”

6) Lines 559-595, conclusions: These could be more clearly stated by referring back to the original research questions found on lines 177-183. As it stands, the reader has to tack back and forth from the conclusions to the questions to see more clearly the degree to which they were answered by the research.

We have made this more clear in the revision, adding in the questions referenced to the conclusion and changing the wording to be clear that we are referring to the research questions in the introduction.

7) I am pleased to see that there was consultation with the Indigenous peoples of the region (lines 225-237). However, the description was also rather vague. Are there organizations of Indigenous peoples who could have been consulted? A better description of the consultative process would be welcome.

We have extended this section to make it clearer. There is not a protocol for consultation at the community level in Argentina or Chile. However, we have strong relationships with many members of the Indigenous communities, and many were interested in this study to understand how the ancient humans relate to them. We communicated our findings to them, including translating our findings into Spanish for them to read. They made many positive comments about the content of the study and found the conclusions very interesting. The ethics section now reads:

Lines 153-165: “During more than 30 years of archaeological work in the region, one of the authors (RAG) held numerous meetings with different members of present-day Patagonian communities living in the same geographic areas where the ancient samples were located. Reaching spaces for dialogue and joint learning between members of the Native and scientific communities has been a central theme since the beginning of his research and allowed exchanging perspectives on the objectives of bioanthropological studies. In addition, on several occasions, RAG organized and carried out educational activities in schools at different educational levels in the cities of Río Grande and Ushuaia, Tierra del Fuego (Argentina). Some members of Native communities expressed interest in establishing through genetic analyses how the ancient humans found by chance and conserved in the museums relate to present-day people. This study followed that interest, and to facilitate the distribution and understanding of our findings, we translated the abstract and the main conclusions into Spanish (see Supplement) and shared them with members of the Native communities.”

Reviewer #2 (Remarks to the Author):

This is a very interesting manuscript that provides a refreshing overview of the population history of Southern Patagonia. There are only a few suggestions or questions:

1. Use of Aónikenk. While there is usually more than one known name for referring to the native groups from Patagonia, the one that present-day communities use the most or prefer should be used. In this particular case is Tehuelche, but along the text the authors used Aónikenk. Is there any justification for this? Otherwise, please change by Tehuelche through the manuscript.

We thank the referee for this suggestion, but on reflection have decided to retain the term Aónikenk. With its broad geographical range, the Tehuelche complex can be divided into several groups. The Tehuelche people living in the South of the Continent (the site of our one sample from the site of Cerro Johnny) are also known as Aónikenk and we prefer to use that name to discourage generalizing our results to the whole Tehuelche complex.

2. The proper spelling of the northwest maritime group should be Kawésqar (q instead of k).

This is an indigenous term and there is no fully agreed upon spelling in Latin script. We chose the Kawésqar spelling as it is in common usage and it matches the terminology of previous population genetic analyses of this group used in the following studies:

de la Fuente, C., Galimany, J., Kemp, B. M., Judd, K., Reyes, O., & Moraga, M. (2015). Ancient marine hunter-gatherers from Patagonia and Tierra Del Fuego: Diversity and differentiation using uniparentally inherited genetic markers. *American Journal of Physical Anthropology*, 158(4), 719–729. <https://doi.org/10.1002/ajpa.22815>

de la Fuente, C., Ávila-Arcos, M. C., Galimany, J., Carpenter, M. L., Homburger, J. R., Blanco, A., Contreras, P., Dávalos, D. C., Reyes, O., Roman, M. S., Moreno-Estrada, A., Campos, P. F., Eng, C., Huntsman, S., Burchard, E. G., Malaspinas, A. S., Bustamante, C. D., Willerslev, E., Llop, E., ... Moraga, M. (2018). Genomic insights into the origin and diversification of late maritime hunter-gatherers from the Chilean Patagonia. *Proceedings of the National Academy of Sciences of the United States of America*, 115(17), E4006–E4012. <https://doi.org/10.1073/pnas.1715688115>

3. The Ethics Statement is a great section. Only two comments: a) Please include the corresponding permits in the supplementary material; b) the authors suggested some work has been done associated with the present-day communities, but it is not clear if they actually support this particular research. Beyond the governmental permissions, is there any statement from the communities to this research? Please enclosed, clarify or add in the acknowledgment section.

We have extended this section to make it clearer. We do not have a community consent statement, as there is not a protocol for this at the community level in Argentina or Chile. However, we have strong relationships with many members of the Indigenous communities, and many were interested in this study to understand how the ancient humans relate to them. We communicated our findings to them, including translating our findings into Spanish for them to read. They made many positive comments on the content

of the study and found the conclusions very interesting. We have also added additional material to the acknowledgements section to be clear about our process including:

Lines 499-500: We are grateful to the members of Patagonian Native communities who accompanied our work, in particular the Selk'nam, Yagán (Yámana), and the Mapuche-Tehuelche.

We have export permit documents that we can share with the journal if needed, but we would not like to include them in the supplement due to them not being necessary for readers to see and the increased visual clutter when added.

4. Regarding samples: did you screen a larger set of samples in order to produce these new 20 genomes? Please include this information in the Material/Method section. This information can contribute to the understanding of the DNA preservation and feasibility of paleogenomic research in the region.

We added a line in the ancient DNA laboratory work to clarify that all of the ancient individuals screened, to our pleasant surprise, yielded sufficient data to use.

There is a report of a modern Yámana sample (yam013), but the reference is incorrect since the referred paper does seem to have whole-genome shotgun sequencing of present-day individuals (only array data)

In the de la Fuente *et al.*, 2018 paper the yam013 individual was indeed whole-genome shotgun sequenced to ~6x coverage (see Table S1 of that paper). The yam024 individual was also whole-genome shotgun sequenced to a similar coverage in that study, but we were unable to obtain good data from this due to bioinformatic issues in converting between genomic coordinate systems (from hg20 to hg19) that we were unable to resolve.

5. Lines 329-330. Please consider using a different expression for referring to ethnographic names. These are not just labels.

We replaced the words “labeling/labels.”

6. Section "Gene Flow from Northern Patagonia into Southern Patagonia in the Mid to Late Holocene": a) According to the SI, the f_4 results seem to be consistent with Chile_Conchali_700BP having an excess allele sharing with later groups only 50% of the time (14 out of 30 cases, counting also the groups). Is this really a consistently significant signal? Please include a sentence with this fact in the section or justify the use of "consistently significant signal"; b) How can be Chile_Conchali_700BP be modeled (individuals from outside Patagonia). Can this contribute to have a deeper understanding of the source of this new ancestry? c) Does this gene flow correlate with previous inferred events described by Posth *et al.*, 2018 and/or Moreno-Mayar *et al.*, 2018? (maybe not in the time, but in the sort of ancestry/signal)

Thanks for this careful reading. The significant f_4 -statistics provided evidence for genetic interaction between Central Chile and Patagonia, but the non-significant f_4 do not disprove a Chile_Conchali_700BP contribution and instead simply reflect limited statistical power.

In our study, our evidence for a widespread *Chile_Conchali_700BP* contribution comes not only from f_4 -statistics, but also from *qpAdm* analyses and *qpGraph* analyses which show that all Late Holocene groups without exception have this admixture signal. In our revision, we make this chain of inference more clear.

We show in the modeling that *Chile_Conchali_700BP* can be modeled as continuous with *Chile_LosRieles_5100BP* (from very near the same location as *Chile_Conchali_700BP*) to the limits of our resolution (there is no evident ancestry from the Andes or Patagonia that contributed to *Chile_Conchali_700BP* but not to *Chile_LosRieles_5100BP*). It is likely that with more data from that region around Central Chile we would detect additional events, but this will require additional data from ancient individuals from the area.

We performed several analyses trying to relate the *Chile_Conchali_700BP* signal to the California Channel Island-related signal of the Posth *et al.*, 2018 paper (which could relate to the Mixe-related signal of the Moreno-Mayar *et al.*, 2018 paper). The California Channel Island-related signal was found in the *Chile_Conchali_700BP* individuals in the Posth study, and it was found at marginal levels in many of the Late Holocene and later individuals. Thus, it is possible that the *Chile_Conchali_700BP*-related ancestry that arrived in Patagonia also carried some of the California Channel Island-related ancestry, but it was an ambiguous signal (potentially due to it being a very small amount of ancestry), so we decided not to include it in this paper.

7. Figure 3: the model is based on the results of the f_4 test. From the individuals used, it seems that Kawésqar_800BP doesn't have a significant excess of allele sharing with Conchali, compared to earlier individuals. How this affects the model? Can the model be replicated using: a) any other Kawésqar individual or b) removing transitions (to decrease the technical artifact mentioned in the figure). Are these results consistent with the current model?

The model in Figure 3 is an admixture graph created using a semi-automated procedure to look through all possible admixture graphs and test them in *qpGraph* to look for fits (*qpGraph* tests admixture graphs by examining all f_2 , f_3 , and f_4 -statistics based on the population relations specified by the graph). The fact that the automated methods found a model that fits the analyses from f_4 -statistics and *qpAdm* provides additional support for the model. The *Kawésqar_800BP* individuals (it is a combination of the two Kawésqar individuals for which we have available data, one 1120BP and one 570BP) also have excess allele sharing with *Chile_Conchali_700BP* as shown by the brown 50% arrow going into them. This is also shown in *qpAdm* analyses that have Kawésqar as ~50% *Chile_Conchali_700BP*-related ancestry. In the revision we repeated the *qpGraph* analyses with only transversions (the prior analyses were done with transitions in CpG sites removed) and show the new Z-scores in the legends of Figures 3 and Supplementary Figure S5. We now report that indeed, with only transversions the graph without the extra edges has a Z-score of 3.3 (vs. 5.1 when all SNPs except for transitions at CpG sites are considered), providing further support for our hypothesis that those edges are artifacts due to shotgun sequenced, non-UDG-treated samples.

8. Software DATES. Why didn't you use the software for estimating the earlier gene flows? (Conchali for example). Is there a technical limitation? The radiocarbon date of the samples

provides a limit, but having an additional estimate can contribute to a better understanding of the process.

We used *DATES* to estimate the Conchali-related mixture into Selk'nam (the largest sample size we had), but we could not obtain reliable dates (the results are in Supplementary Online Table 6C #4) likely due to the low sample size and the fact that the admixture is older so the power is lower. Our date of $\sim 7,711 \pm 1,417$ BP is older than the ancestral source (LaArcillosa2_5800BP), and the covariance curves do not look reliable, so we did not emphasize this in the manuscript.

9. Line 563: "arrival of a later stream of people". Does it involve the arrival of new people or, as the authors suggested, involve the genetic differentiation of maritime groups maintaining gene flow between them (with lower rates in the last millennium)? Please rephrase the sentence considering that population changes cannot only be explained by the arrival of new people.

We argue in the revised manuscript that these patterns reflect arrival of new people into the region rather than only genetic differentiation over time. The Punta Santa Ana (~6600BP) individual is in the location of most of the later Kawéskar individuals, while the Ayayema ~4700BP individual is further north. When PuntaSantaAna or Arcillosa2 are included as outgroups in *qpAdm* analyses, the models still fit for Late Holocene maritime-diet groups, providing another way to see how ancestry of the sort that was present in western South Patagonia prior to 5000BP was displaced by a different type of ancestry present in the north of South Patagonia around 4700BP. This argument is made more clear in the revision in the Admixture Graph Model and Conclusion sections.

10. The authors briefly mentioned population sizes inferred by the density of archaeological sites. Does the data allow any population size estimates? (using MSMC for example).

This is an extremely interesting question, which we now address in part in this revision. Unfortunately, MSMC requires whole-genome sequenced diploid data, which requires coverages usually over 20x to make accurate diploid calls (the highest coverage we have is about 10x). In addition, it is generally not reliable for estimates within the past few thousands of years, so it would probably not work here. However, we used conditional heterozygosity, which is one measure of population size, and now show that ancient Patagonians had lower diversities than individuals from the Andes, and indeed had diversities comparable to those of the least diverse human populations today (from Amazonia). This method requires at least 2 individuals, though, so we could not test for changes in population size over time within Patagonia, because our individuals over 2000 years old are single individuals from different sites, so grouping them would lead to an upward bias relative to groups where all individuals are from the same site.

We added the Conditional Heterozygosity analyses in the manuscript and discuss the limitations. We also clarify:

Lines 199-201: "We were not able to determine the date of the population bottlenecks that produced this low variation, because the three Middle Holocene Patagonians were from different sites, so there might be an upward bias when we grouped them. Moreover,

higher resolution reconstruction of population size change over time requires high coverage whole genome sequencing data, which we do not have."

Reviewer #3 (Remarks to the Author):

This is a very interesting paper and the authors are very thorough with the population genetic analyses utilized to form their conclusions. My main comments refer to the structure of the manuscript, including a missed opportunity to highlight how these samples relate to other ancient genomes available from South America. I am also concerned that many of the interesting results have been buried in the supplement. That being said, here are my suggested edits:

1. The abstract reads like a results section. I would suggest rewording to a summary fashion. The first two sentences are especially descriptive.

We have shortened the abstract to be more of a summary.

2. For the introduction, while I appreciate the in-depth coverage of the cultures, the authors may want to summarize with some key differences between the five populations and move the rest to the supplement.

We have shortened the introduction substantially and moved much of the material to the supplement.

2. I was especially pleased to see that the ethics statement was included in the main manuscript. However, the description of where the samples are from should be sent to the supplement and perhaps keep the statement about community engagement with indigenous communities in the manuscript.

We have now made this suggested change.

3. On the other hand, some of the analyses performed are quite interesting but they are buried in the supplement. For examples, the comparisons between linguistic, geographic, and temporal distances with genetic drift seems very interesting and novel. Perhaps the authors could visualize the results in some manner and put it in the main manuscript. Conversely, I don't particularly find Figure 2 to be informative or interesting.

On reflection we have chosen to retain Figure 2, which presents many of the qualitative results in easily accessible form. It shows clearly the genetic cline following the coast among the Late Holocene samples. It also shows that the Ayayema sample is closer to the Kawéskar and Yámana individuals than to the others, Middle Holocene samples considered. Both are major results of our study. Both panels of the figure make it clear how individuals from the same geographical group tend cluster together genetically.

We agree that Mantel test correlations could be better highlighted, and added a Main Table for this. There are ways to plot Mantel test correlations, but they usually are not the

most visually attractive (and it is particularly difficult with multiple variables at once that are correlated to each other). We could show the correlations after correcting for the other variables, but the visual would not add significantly beyond stating the result.

3. It is strange to be referring to the ancient samples by historical ethnic groups for all of the reasons mentioned by the authors. Wouldn't it be less problematic to refer to the samples by the region from which they were unearthed?

We extensively discussed this amongst ourselves and re-opened the debate on it among our co-authors after receiving the reviewer's comments on this point. On reflection, we decided to continue to refer to the Late Holocene ancient individuals by the historical ethnic groups from the locations where the ancient individuals were found primarily due to greater manuscript readability. Our rationale here is that it would be quite difficult for readers to keep track not only of the regional areas but also of which ethnic groups lived in each one. The geographic region names are also often harder to remember than the ethnic group names. We believe the Mantel tests justified our naming of the ancient individuals by the ethnic groups that existed at European contact in those regions, and now explain this.

4. It seems that the authors draw many conclusions from the F4 analyses. I would recommend weighing all of the evidence from the plethora of analyses done before drawing such hefty conclusions.

Our approach in this paper was first to use more qualitative analyses, such as uniparental markers and clustering methods (ADMIXTURE, PCA, MDS, and a neighbor-joining tree), and then to progress to more formal statistical tests, letting the evidence build gradually to support our main conclusions. The qualitative analyses provided suggestive evidence for almost all of our main conclusions. For example, the fact that the PuntaSantaAna and Arcillosa2 individuals are genetically a clade with respect to more recent Patagonians, despite their having maritime and terrestrial diets, respectively, can be seen in the MDS plot, where they cluster together (and they show similar ADMIXTURE patterns even up to $K=7$). The significance of this signal was then demonstrated rigorously with f_4 -statistics. The same occurred with the finding that the Ayayema individual has ancestry found specifically in Late Holocene maritime-diet individuals, which can be seen in the MDS plot (and also partially in the ADMIXTURE plot), but rigorously shown with f_4 -statistics. We also then followed this up with *qpAdm* analyses and *qpGraph* analyses, which showed more clearly that the PuntaSantaAna/Arcillosa2 ancestry was replaced on the west coast in the maritime-diet individuals by the Ayayema-related ancestry. The finding about the mixtures between the groups could be seen partially in the MDS plot and neighbor-joining tree, but needed to be more rigorously shown with f_4 -statistics, *qpAdm*, and *DATES*. The discovery about *Chile_Conchali_700BP* ancestry was suggested by the f_4 -statistics and then shown to be consistent across all populations through *qpAdm* and *qpGraph* analyses. The analyses relating the present-day individuals to the ancient individuals was done only with *qpGraph*, because that allowed us to best incorporate the European ancestry without biasing the analysis of the indigenous ancestry.

5. I also think that at least one of the analyses with regard to continental gene flow should be visualized and featured in the manuscript. For example, was the admixture graph tried with other available genomes from South America? I think this would be a very interesting result to see. It would also be nice to see the continental level analysis presented in a PCA or structure plots. I realize that this is a regional paper but given that such few ancient genomes are available for South America, I believe that considering the samples in a larger scope would increase the impact of the paper.

We were and are still extremely interested in the relationship between Patagonia and the rest of South America. We found that Patagonia was essentially a clade with respect to all of the currently published ancient South Americans except for the *Chile_Conchali_700BP* individuals (who are a clade with the *Chile_LosRieles_5100BP* individual with respect to all other ancient South Americans). Thus, our only finding of interaction between Patagonia and the rest of South America was ancestry coming from Central Chile (related to *Chile_Conchali_700BP*) into South Patagonia after ~4700BP. The rest of the ancient South Americans are outgroups relative to the Patagonians (so when they are put in the admixture graph, they are in the graph in a position right after the *USR1* individual). Figures S1 and S2 have all the ancient South Americans previously published, though no patterns are obvious. Thus, the present manuscript does not add significantly to our knowledge about deep South American population history, and we chose not to attempt to engage those questions in this study in our final manuscript.

REVIEWERS' COMMENTS:

Reviewer #2 (Remarks to the Author):

I appreciate the changes and explanations provided by the authors. The revised version of the manuscript targets all the issues/questions of the reviewers and there are no new comments. For the sake of transparency, I would stress on the idea of sharing the corresponding export permit documents as other papers have done (e.g. Moreno-Mayar et al., 2019)

Reviewer #3 (Remarks to the Author):

The authors have satisfactorily addressed my comments and concerns.